# Presence of *Campylobacter*
*jejuni* and *C. coli* in Dogs under Training for Animal-Assisted Therapies

**DOI:** 10.3390/ijerph18073717

**Published:** 2021-04-02

**Authors:** Antonio Santaniello, Lorena Varriale, Ludovico Dipineto, Luca Borrelli, Antonino Pace, Alessandro Fioretti, Lucia Francesca Menna

**Affiliations:** 1Departments of Veterinary Medicine and Animal Productions, Federico II University of Naples, 80134 Naples, Italy; lorena.varriale@unina.it (L.V.); ludovico.dipineto@unina.it (L.D.); luca.borrelli@unina.it (L.B.); antonino.pace@unina.it (A.P.); alessandro.fioretti@unina.it (A.F.); luciafrancesca.menna@unina.it (L.F.M.); 2Marine Turtle Research Centre, Stazione Zoologica Anton Dohrn, 80055 Portici, Italy

**Keywords:** animal-assisted therapies, co-therapist dogs, *Campylobacter* spp., dog training center, survey, zoonoses, public health

## Abstract

This study was conducted to evaluate the presence of *Campylobacter* (C.) *jejuni* and *C. coli* in dogs at five dog training centers in Southern Italy. A total of 550 animals were sampled by collecting rectal swabs. The samples were processed to detect thermotolerant *Campylobacter* spp. by culture and molecular methods. *Campylobacter* spp. were isolated from 135/550 (24.5–95% confidence interval) dogs. A total of 84 *C. jejuni* (62.2%) and 51 *C. coli* (37.8%) isolates were identified using conventional PCR. The dog data (age, sex, breed, and eating habits) were examined by two statistical analyses using the *C. jejuni* and *C. coli* status (positive or negative) as dependent variables. Dogs fed home-cooked food showed a higher risk of being positive for *C. jejuni* than dogs fed dry or canned meat for dogs (50.0%; *p* < 0.01). Moreover, purebred dogs had a significantly higher risk than crossbred dogs for *C. coli* positivity (16.4%; *p* < 0.01). This is the first study on the prevalence of *C. jejuni* and *C. coli* in dogs frequenting dog training centers for animal-assisted therapies (AATs). Our findings emphasize the potential zoonotic risk for patients and users involved in AATs settings and highlight the need to carry out ad hoc health checks and to pay attention to the choice of the dog, as well as eating habits, in order to minimize the risk of infection.

## 1. Introduction

Dogs are playing an increasing role as supporters or co-therapists for people with psychological or physical disabilities [1]. The benefits of interaction with dogs in the healthcare context mainly consist of outcomes from animal-assisted therapies (AATs), defined as a therapeutic intervention incorporating animals to improve health and wellness [2]. In particular, AATs with dogs represent non-pharmacological therapies or co-therapies to support psychotherapy or other therapies [3,4,5,6,7]. AAT interventions have been applied to address different kinds of illness, including adults and children with autism spectrum disorder [8,9], people with Alzheimer’s disease and other dementias [10,11,12], and during psychotherapy for adolescents [13].

Dogs are the main animal species involved in AATs [5,14,15], but despite the benefits derived from their competence and their interspecific relationship with humans, this animal species could represent a vector of several zoonotic agents’ transmission [16,17,18,19,20,21,22]. As reported by Ghasemzadeh and Namazi [22], dogs are a major reservoir of zoonotic infections and they can transmit several viral and bacterial diseases to humans. Zoonotic bacterial diseases can be transmitted to humans by infected saliva, aerosols, contaminated urine or feces, and by direct contact with the dog. Such infections are due to different zoonotic agents such as *Pasteurella multocida, Salmonella* spp., *Brucella canis, Yersinia enterocolitica, Capnocytophaga canimorsus, Bordetella bronchiseptica, Coxiella burnetii, Leptospira* spp., *Staphylococcus intermedius*, and Methicillin-resistant *Staphylococcus aureus* (MRSA), including *Campylobacter* spp., and particularly *C. jejuni* and *C. coli* [22]. These *Campylobacter* species are the most frequent bacterial cause of acute human gastroenteritis in many industrialized countries [23]. *C. jejuni* and *C. coli* can be commensal inhabitants in the intestinal tract of many mammals and avian species [24,25]. It was reported that the main risk factors for humans include the consumption of contaminated food (mainly poultry meat) and drinking water, but direct contact with carrier animals was also found to be a possible source of infection of *C. jejuni* and *C. coli* [26,27,28,29,30,31,32]. Dogs can be healthy carriers of *Campylobacter* spp., showing higher carriage rates in the case of animals under six months of age with or without diarrhea, whereas in older dogs, no difference in *Campylobacter* spp. shedding was reported between healthy and diseased animals [33].

As reported above, although several studies were carried out to assess the presence of *Campylobacter* spp. in dogs, no previous research considered dogs involved in AATs. Therefore, the aim of our study was to evaluate the presence of *Campylobacter* spp. (i.e., *C. jejuni* and *C. coli*) in dogs frequenting dog training centers in Southern Italy for AATs.

This study enriches the international scientific literature regarding the potential risks of transmission of *C. jejuni* and *C. coli* by dogs involved in AATs, underlining the need to expand health protocols and related hygiene practices for AATs, thus guaranteeing the health of patients and the safety of care.

## 2. Materials and Methods

### 2.1. Sampling

Our study was undertaken between October 2018 and May 2019 in five dog training centers located in Southern Italy. Rectal swab samples were collected from a total of 550 dogs. This sample size was calculated using the formula proposed by Thrusfield [34] for a large (theoretically infinite) population using the following values: expected prevalence (8.0%), confidence interval (95%), and desired absolute precision (5%). The dog training centers were identified and named as C1, C2, C3, C4, and C5, in which 180, 112, 180, 35 and 43 dogs were housed, respectively, and each one was sampled. Each dog was apparently in good health and was individually sampled using rectal swabs. The information for each dog was collected through an interview performed on arrival at the dog training center by researchers of the working group using a semi-structured questionnaire addressing some generic characteristics (age, sex, breed, and eating habits) and different questions regarding health status. The dogs were classified into two age groups: one containing animals from three to six months of age (*n* = 245) and the other containing animals older than six months (*n* = 305); two sex groups, male (*n* = 299) and female (*n* = 251); two breed groups, crossbred (*n* = 385) and purebred (*n* = 165); and three eating habits groups, dry dog food (*n* = 378), canned meat for dogs (*n* = 154), and home-cooked food (*n* = 18).

### 2.2. Bacterial Isolation

The rectal swab samples were stored in Amies Transport Medium (Oxoid, Basingstoke, UK) at 4 °C, and transported to the laboratory and analyzed within 2 h of collection. Samples were inoculated into Campylobacter-selective enrichment broth (Oxoid, Basingstoke, UK) and incubated at 42 °C for 48 h under microaerobic conditions provided by CampyGen (Oxoid, Basingstoke, UK). Subsequently, each sample was streaked onto Campylobacter blood-free selective agar (CCDA; Oxoid, Basingstoke, UK) with the corresponding supplement (SE 155; Oxoid, Basingstoke, UK). After incubation at 42 °C for 48 h under microaerobic conditions, the plates were examined for typical *Campylobacter* colonies. From each suspected plate, a loopful of colonies was purified on sheep blood agar (SBA; Oxoid, Basingstoke, UK) and finally incubated for 24 h at 42 °C under microaerobic conditions. The colonies comprising curved or spiral motile rods, when observed under phase contrast microscopy, were presumptively identified as *Campylobacter* spp. and then identified at the species level by reaction to Gram staining; oxidase, catalase, and hippurate tests; as well as susceptibility to nalidixic acid and cephalothin, according to the International Standard Procedures [35].

### 2.3. Polymerase Chain Reaction (PCR)

The extraction and purification of DNA from isolated colonies on sheep blood agar was performed using a Bactozol kit (Molecular Research Centre, Inc., Cincinnati, OH, USA) as described previously [36]. The specific detection of the *Campylobacter* genus was based on PCR amplification of the *cadF* gene using the oligonucleotide primers cadF2B and cadR1B, as described by Santaniello et al. [24].

All DNA extracts were also examined by duplex PCR for the presence of *C. jejuni* and *C. coli* species using amplification conditions and the oligonucleotide primers ICJ-UP and ICJ-DN, ICC-UP, and ICC-DN, as previously described [36]. PCR products were separated by electrophoresis on 1.5% agarose gels (Gibco–BRL, Milan, Italy), stained with ethidium bromide, and visualized under UV light. PCR amplified without the DNA template was used as the negative control, whereas two reference *Campylobacter* strains, *C. jejuni* ATCC 29428 and *C. coli* ATCC 33559, obtained from LGC Promochem (LGC Promochem, Teddington, UK) were used as positive controls.

### 2.4. Data Analysis

All the statistical analyses were performed using SPSS 20 Software for Microsoft Windows (SPSS Inc., Chicago, IL, USA). Data were recorded in an Excel file. The dog data (age, sex, breed, and usual food) underwent univariate analysis (Pearson’s chi-squared test for independence) using the *C. jejuni* and *C. coli* status (positive or negative) as dependent variables. Only the independent variables that showed significant differences (*p* < 0.05) in the univariate test were used for the logistic regression model. If interaction between variables was suspected, a logistic regression model was run with and without these variables to evaluate possible effect modification [37].

## 3. Results

Out of the 550 dogs examined, 135 (24.5%; 95% confidence interval (CI) = 21.0–28.4%) were positive for *Campylobacter* spp. As shown by PCR, 84/135 (62.2%, CI = 53.4–70.3%) positive samples were identified as *C. jejuni*, whereas 51/135 (37.8, CI = 29.7–46.6%) positive samples were identified as *C. coli.* Particularly, dogs fed home-cooked food showed a high prevalence of *C. jejuni,* at 50.0%, whereas dogs fed with dry dog food and canned meat for dogs showed a prevalence of 14.3% and 13.7%, respectively. These differences were statistically significant (*p* < 0.01), as shown in Table 1. Purebred dogs showed a prevalence of 16.4% (95% CI = 11.2–23.1%) for *C. coli*, whereas crossbred dogs showed a prevalence of 6.2% (95% CI = 4.1–9.3%); this difference was statistically significant (*p* < 0.01), as shown in Table 2. In contrast, there was no significant difference related to age and sex (*p* < 0.05). With respect to the statistical regression model results, breed and eating habits were risk factors for *C. coli* and *C. jejuni* positivity, respectively. Specifically, purebred dogs had a significantly higher risk of being positive for *C. coli* than crossbred dogs (odds ratio (OR) = 2.042; *p* < 0.01). Dogs fed home-cooked food had a significantly higher risk of carrying *C. jejuni* than dogs fed with canned meat (OR = 4.766; *p* = 0.002) and dogs fed with dry food (OR = 3.831; *p* = 0.006). All results of the logistic regression model are listed in Table 3.

Given that the dog training centers examined were different in management and geographic location and were sampled at different times, statistical analysis within each center was conducted, considering the same group categories (age, sex, breed, and usual food) analyzed on the total number of examined dogs.

In C1, out of the 180 dogs examined, a total of 34 (18.9%; 95% CI = 13.6–25.5%) were positive for *Campylobacter* spp. As determined by PCR, 24 (13.3%) were positive for *C. jejuni* and 10 (5.5%) for *C. coli*. Purebred dogs (35.0%) showed a *C. jejuni* prevalence of 20.6% (95% CI = 11.9–33.0%), whereas crossbred dogs (65.0%) showed a prevalence of 9.4% (95% CI = 5.0–16.6%); this difference was statistically significant (*p* < 0.05).

In C2, out of the 112 dogs examined, a total of 21 (18.7%; 95% CI = 12.2–27.5%) were positive for *Campylobacter* spp. As determined by PCR, 13 (11.6%) were positive for *C. jejuni* and 10 (8.9%) for *C. coli,* but 2 dogs were positive for both *C. jejuni* and *C. coli* at the same time. Purebred dogs (32.1%) showed a *C. jejuni* prevalence of 22.2% (95% CI = 10.7–39.6%), whereas crossbred dogs (67.8%) showed a prevalence of 6.6% (95% CI = 2.4–15.3%); this difference was statistically significant (*p* < 0.05). In addition, purebred dogs showed a *C. coli* prevalence of 19.4% (95% CI = 8.8–36.6%), whereas crossbred dogs showed a prevalence of 3.9% (95% CI = 1.0–11.9%); this difference was statistically significant (*p* < 0.01).

In C3, out of the 180 dogs examined, a total of 46 (25.6%; 95% CI = 19.5– 32.7%) were positive for *Campylobacter* spp. As determined by PCR, 26 (14.44%) were positive for *C. jejuni* and 20 (11.11%) for *C. coli*. Purebred dogs (24.4%) showed a *C. coli* prevalence of 22.7% (95% CI = 12.0–38.2%)*,* while crossbred dogs (75.5%) showed a prevalence of 11.03% (95% CI = 6.51–17.83%); this difference was statistically significant (*p* < 0.05). In this dog training center, the 92 dogs under six months of age showed a higher prevalence of *C. coli* (19.3, 95% CI = 12.0–29.4%) compared with the 88 dogs older than six months, and this difference was statistically significant. In addition, although these results were conditioned by the small size of the samples, the dogs fed home-cooked food showed a very high prevalence of *C. jejuni* (100%; 95% CI = 31.0–96.8), with a statistically significant difference compared with dogs fed with dry dog food and canned meat for dogs (*p* < 0.05).

In C4, out of the 35 dogs examined, a total of 18 (51.4%; 95% CI = 34.3–68.3%) were positive for *Campylobacter* spp. As determined by PCR, 12 (34.3%) were positive for *C. jejuni* and 6 (17.1%) for *C. coli*. The data from this dog training center showed no statistically significant differences.

In C5, out of the 43 dogs examined, a total of 16 (37.2%; 95% CI = 23.4–53.3%) were positive for *Campylobacter* spp. As determined by PCR, 10 (21.7%) were positive for *C. jejuni* and 6 (13.9%) for *C. coli*. In this dog training center, the 18 dogs under six months of age showed a higher prevalence of *C. jejuni* (66.7, 95% CI = 41.1–85.6%) than the 25 dogs older than six months, and the difference was statistically significant.

## 4. Discussion

Dogs are the main animal species chosen for AATs [2,5,15]. They are keen observers of human reactions through their exceptional ability to read signs of will and emotion from human faces [38]. In addition, dogs can read the non-verbal language of humans [39,40,41], probably deriving from the history of coevolution with human beings, the ethogram, and the breed [5].

Usually, AATs are performed in healthcare facilities and are prescribed to patients with different illnesses belonging to risk categories (e.g., dialysis, hospitalized, and immunosuppressed or immunocompromised) [42,43,44]. Patients interact with dogs through different activities (i.e., petting, brushing, leading on a leash, hiding a ball, etc.) [5].

As reported by Shen et al. [45] in their recent systematic review, bodily contact with the animal was the primary factor with respect to the other themes identified as facilitators of effectiveness in these interventions. During these activities, because of repeated contact with the dog’s body and mucosae, involved patients could be exposed to zoonotic pathogens (e.g., bacteria, viruses, and fungi) potentially transmitted by the dog through direct contact [16,17,18,20,21,22]. As dogs have been reported to be carriers of *Campylobacter* spp. [46,47,48], their potential role in the transmission of this pathogen to humans should not be underestimated from a public health perspective. Thermotolerant *Campylobacter* species represent the main cause of human gastroenteritis in Europe [23] and, although it occurs mainly as a foodborne disease, about 6% of human campylobacteriosis is linked to contact with pets [49]. To the best of our knowledge, this is the first study assessing the prevalence of *Campylobacter* spp. in dogs involved in AATs and our results contribute to focusing on an interesting topic of public health.

The findings of this survey demonstrate the occurrence of *Campylobacter* spp. in dogs at all five centers examined (24.5%), with a prevalence of 15.3% for *C. jejuni* and 9.3% for *C. coli*. The overall presence of *Campylobacter* spp. showed in the present study does not completely reflect the results of previous research, where the prevalence of *Campylobacter* spp. ranged between 4.8% and 75.8% [50,51,52,53]. This wide range may be linked to differences in the populations examined, as well as to the identification methods used. *C. jejuni* has been reported as the most predominant species, whereas *C. coli* showed mostly lower rates. Compared with our results, Thèpault et al. [54] and Karama et al. [55] reported a higher prevalence of *C. jejuni* (24.4% and 29.1%, respectively), but lower values regarding *C. coli* (2.6% and 5.4%, respectively). In Slovakia, Badlìk et al. [56] isolated *C. jejuni* and *C. coli* with a prevalence of 51.2% and 9.8%, respectively. Wieland et al. [31] isolated *C. jejuni* with a prevalence of 5.7% and *C. coli* with a prevalence of 1.1%, whereas in Norway, Sandberg et al. [57] isolated *C. jejuni* with a prevalence of 3.0%. In Italy, Rossi et al. [58] conducted a survey in dogs and cats, isolating *Campylobacter jejuni* in 8.9% of 190 dogs sampled.

Interestingly, in our study, breed and dog feeding were the risk factors significantly associated with *Campylobacter* spp. occurrence, while differences in age and sex were not statistically significant. In agreement with our findings, Ahmed et al. [59] reported sex and age as risk factors with no statistically significant association with *Campylobacter* culture positivity, whereas breed, health status, and cohabitation with other dogs had a statistically significant association. Many studies demonstrated that younger dogs were more likely to harbor *Campylobacter* spp., probably due to an immature immune system and an underdeveloped gut microbiota that is unable to perform the competitive exclusion toward pathogens [55]. Although breed has not been reported as a risk factor for *Campylobacter* spp. occurrence in dogs, our findings suggest that purebred dogs are more susceptible to *Campylobacter* spp. colonization compared with crossbred dogs, which are generally more resistant to disease. Although it was not possible to speculate on the highest prevalence of *C. coli* in purebred dogs, we hypothesize that the strong selection for morphological characteristics in these animals may influence susceptibility to infection. In addition, with respect to food, it is recognized that homemade cooked food, especially meat, may represent a source of *Campylobacter* spp. and a potential risk factor for dogs [60]. This finding is supported by our study showing increased prevalence of *C. jejuni* in dogs fed home-cooked food, although further investigation is needed to understand if this is linked to poor food handling practices or direct exposure from raw food.

Future studies regarding the assessment and analysis of the risk of transmission of *Campylobacter* should consider the time of exposure to the pathogen, the kind of patients involved, the setting, as well as the modalities of interaction between dog and patient. Particularly, contact and time of exposure could represent very important factors of transmission, since the duration of an AAT intervention can range from 15 to 120 min [45]. During the cycle of interventions, the interactions between dog and patient become increasingly close and intense due to the intensification of the interspecific relationship [5]. In addition, further studies are needed to understand the mechanisms underlying the variable-related differences reported in this study.

Finally, considering the few data and more generic guidelines often referenced in the scientific literature, our findings serve to enrich the general recommendations for the health control of dogs and related risk assessment in the field of AATs [44,61,62,63,64].

### Limitations

While this study shows some interesting results, it also has some limitations. Although sampling was not carried out in the colder or warmer months of the year during the study period, seasonality was not considered as a variable that could influence the results. Further, in this study, the presence of other *Campylobacter* species, such as *C. upsaliensis*, *helveticus*, and *C. lari*, was not evaluated. The antibiotic sensitivity of the isolates was not evaluated. A single swab was collected from each dog, but multiple parts of an animal’s body could be checked. Finally, no samples were taken from the cooked foods eaten by the dogs who tested positive.

## 5. Conclusions

The intent of our study is not to limit the participation of dogs in future AATs but to highlight the importance of performing health checks on dogs involved in these types of interventions and to prevent the risk of transmission of pathogens such as *C. jejuni* and *C. coli*. Public health concerns, such as zoonoses, are actually tackled through a multidisciplinary and integrated One Health approach. This new strategy encompasses collaborative actions to improve surveillance, prevent and control infection and relay key messages to public and professional audiences. In light of this concept, our study was conceptualized to focus on the specific setting of vulnerable people and animals in contact with each other. As widely reported in the literature, AATs are aimed exclusively at people with various types of diseases and sometimes immunocompromized or immunosuppressed conditions. Therefore, an adequate sanitary monitoring protocol that includes different zoonotic agents (bacteria, viruses, and parasites), as well as good dog management practices are essential to protect both the health of the patients involved and the welfare of the animals.

## Figures and Tables

**Table 1 ijerph-18-03717-t001:** Dog data and positivity for *Campylobacter jejuni*.

Dog Data	No. of Tested Dogs	No. of Positive Dogs	%	95% CI	*p* *
Age					
<6 months	245	41	16.7	12.4–22.1	0.393
>6 months	305	43	14.1	10.5–18.6	
Sex					
Male	299	42	14.0	10.4–18.6	0.383
Female	251	42	16.7	12.4–22.1	
Breed					
Crossbred	385	53	13.8	10.6–17.7	0.134
Purebred	165	31	18.8	13.3–25.8	
Eating habits					
Dry food	378	54	14.3	11.0–18.3	0.000
Canned meat	154	21	13.6	8.8–20.3	
Home-cooked	18	9	50.0	26.8–73.2	
Total	550	84	15.27	12.42–18.62	

CI, confidence interval; * Chi-square.

**Table 2 ijerph-18-03717-t002:** Dog data and positivity for *Campylobacter coli*.

Dog Data	No. of Tested Dogs	No. of Positive Dogs	%	95% CI	*p* *
Age					
<6 months	245	26	10.6	7.2–15.3	0.332
>6 months	305	25	8.2	5.5–12.0	
Sex					
Male	299	30	10.0	7.0–14.1	0.502
Female	251	21	8.4	5.4–12.7	
Breed					
Crossbred	385	24	6.2	4.1–9.3	0.000
Purebred	165	27	16.4	11.2–23.1	
Eating habits					
Dry food	378	39	10.3	7.5–13.9	0.446
Canned meat	154	11	7.1	3.8–12.7	
Home-cooked	18	1	5.6	0.3–29.4	
Total	550	51	9.3	7.0–12.1	

CI, confidence interval; * Chi-square.

**Table 3 ijerph-18-03717-t003:** Results of logistic regression model.

Independent Variable	Standard Error	*p* Value	Odds Ratio	95% Confidence Interval	
				Low	High
Breed *					
Purebred vs. Crossbred	0.210	0.001	2.042	1.352	3.084
Eating habits **					
Dry food vs. Canned meat	0.232	0.346	0.804	0.510	1.266
Dry food vs. Home-cooked food	0.489	0.006	3.831	1.469	9.992
Canned meat vs. Home-cooked food	0.514	0.002	4.766	1.739	13.057

Dependent variable is *Campylobacter jejuni* ** or *Campylobacter coli* * positivity.

## Data Availability

Data are available upon request to the corresponding author.

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
