# Peer review of "Presence of Campylobacterjejuni and C. coli in Dogs under Training for Animal-Assisted Therapies"

_ijerph, 2021, doi:10.3390/ijerph18073717_

Round 1
Reviewer 1 Report
Revision of manuscript ijerph-1151328
Dear Authors,
Your manuscript entitled “Presence of Campylobacter jejuni and C. coli in dogs under training for Animal Assisted Therapies” is an interesting survey on the presence of zoonotic Campylobacter in feces of dogs employed for “humans therapy”. The work was well planned and conducted, and it provided valuable information on the zoonotic risk patient could be potentially exposed to. This kind of study are highly appreciated, in my opinion.
However, I found some inaccuracies and errors in the manuscript, see below, that must be solved before a full positive evaluation.
- Material and Methods:
- Lines 84-88: all these data are results; please, evaluate the possibility to move all the number in results section.
- Line 92: please better specify the enrichment broth employed.
- Line 107: “cadF” in italic.
- Lines 114-115: Authors speak about 3 reference strains, but only 2 were listed.
- Results:
- Lines 160-161: this is a very interesting results, but it opens some questions: 1) why did not this datum emerge before? Authors reported that among 135 positive samples 84 were jejuni while 51 were C. coli; these data did not match together, indeed if you add positive dogs from all the 5 centers the total number is 138; please clarify; 2) How many colonies for each samples were selected on CCDA to be tested for genus and species? Please add this important information to material and method section.
- Lines 179-182: “total of 21 (60.0%; 95% CI = 42.2 – 75.6%) were positive for Campylobacter spp. As determined by PCR, 12 (34.3%) were positive for C. jejuni and 6 (17.1%) for C. coli.” Campylobacter species data is missing for 3 isolates.
- Line 183: 16 instead 46.
- Lines 154-187: Total number of positive dogs for campylobacter emerging from this part is 138 that is in contrast with the results previously reported (“135 [24.5%; 95% confidence interval (CI) = 21.0 – 28.4%] were positive for Campylobacter spp” lines 127-128), furthermore, 85 were jejuni and 52 were C. coli (85+52 =137), that is in contrast with data reported previously; all these number did not match, please correct.
- Discussion
- Lines 189-215Some of these information are already reported in introduction; furthermore, Authors are invited to focus on discussion of obtained results with few digressions. A brief introduction to discussions is acceptable and significant, but, in my opinion, in this case, it is too long.
- A dipper discussion on obtained results should be appreciated, but I have no specific suggestion about this.
- The situation emerged considering all data together is not the same considering the single centers, if I well understand; this should be discussed if possible.
- Considering that One Health is an “hot topic” in this period and considering this work exactly fits with this concept, some consideration (here or in conclusion section) should be provided.
I sincerely hope that these suggestions will enhance this manuscript. However, if I have made any errors or misinterpretations, I apologize in advance.
I am sorry for the delay in the revision
Sincerely
The Reviewer
Author Response
Please, see the attachment.

Reviewer 2 Report
The manuscript entitled “Presence of Campylobacter jejuni and C. coli in dogs under training for Animal Assisted Therapies” is an interesting study that may be relevant for many people working or living with dogs. The role of dogs as co-therapists for psychological or physical disabilities in humans can be very helpful.
I have some minor points that the authors may consider:
Abstract:
Line 13: Campylobacter (C.) jejuni.
Lines 17-18: A total 84 C. jejuni (62.2%) and 51 C. coli (37.8%) isolates were identified using conventional PCR
Introduction:
Line 34: contributes to instead of consist in.
Line 54: C. jejuni and C. coli can be commensal inhabitants in the intestinal tract.
Line 56: Please add (mainly) before poultry meat, as poultry meat is the main but not solely.
Materials and Methods:
Line 78: The authors should clarify the difference in numbers of collected samples between the five centers: is it a constant percentage in each center or the samples were taken from all animals in each center. Is the number of samples related to the number of animal capacity in each center?
Line 98: and finally incubated for 24 h at 42°C under microaerobic conditions?
C. upsaliensis and lari are relevant species among dogs (Acke et al., 2009; Leonard et al., 2011; Parsons et al., 2010, 2011; Procter et al., 2014; Acke et al., 2018). C. upsaliensis can cause enteric disease also in humans (Labarca et al., 2002). Dogs are known to be key reservoirs of C. upsaliensis in addition to C. helveticus (Acke et al., 2018).
It seems the authors have not considered these Campylobacter species in this study, so they should explain why and maybe mention it as a limitation.
Line 114: Only two reference strains are mentioned.
Line 119: recorded in an Excel file.
Results:
Please check the numbers:
Line 179: The authors mentioned in sampling that collected samples from C4 were 43 not 35.
Line 183: The authors mentioned in sampling that collected samples from C5 were 35 not 43.
Line 183: In C5, out of the 43 dogs examined, a total of 46 ?? ...
Author Response
Please, see the attachment.

Round 2
Reviewer 1 Report
Dear Authors,
You modified the manuscript and/or answered way to my questions in a satisfactory.
I have no more queries.
Good work.